# Estimating the Growing Stem Volume of the Planted Forest Using the General Linear Model and Time Series Quad-Polarimetric SAR Images

**DOI:** 10.3390/s20143957

**Published:** 2020-07-16

**Authors:** Jiangping Long, Hui Lin, Guangxing Wang, Hua Sun, Enping Yan

**Affiliations:** 1Key Laboratory of Forestry Remote Sensing Based Big Data & Ecological Security for Hunan Province, Changsha 410004, China; longjiangping@csuft.edu.cn (J.L.); gxwang@siu.edu (G.W.); sunhua@csuft.edu.cn (H.S.); enpingyan@csuft.edu.cn (E.Y.); 2Key Laboratory of State Forestry Administration on Forest Resources Management and Monitoring in Southern Area, Changsha 410004, China; 3Department of Surveying Engineering, College of Civil Engineering, Central South University of Forestry and Technology, Changsha 410004, China; 4Department of Geography and Environmental Resources, Southern Illinois University, Carbondale, IL 62901, USA

**Keywords:** growing stem volume, general linear model, polarimetric SAR, Yamaguchi decomposition, planted forest

## Abstract

Increasing the area of planted forests is rather important for compensation the loss of natural forests and slowing down the global warming. Forest growing stem volume (GSV) is a key indicator for monitoring and evaluating the quality of planted forest. To improve the accuracy of planted forest GSV located in south China, four L-band ALOS PALSAR-2 quad-polarimetric synthetic aperture radar (SAR) images were acquired from June to September with short intervals. Polarimetric characteristics (un-fused and fused) derived by the Yamaguchi decomposition from time series SAR images with different intervals were considered as independent variables for the GSV estimation. Then, the general linear model (GLM) obeyed the exponential distribution were proposed to retrieve the stand-level GSV in plantation. The results show that the un-fused power of double bounce scatters and four fused variables derived from single SAR image is highly sensitive to the GSV, and these polarimeric characteristics derived from the time series images more significantly contribute to improved estimation of GSV. Moreover, compared with the estimated GSV using the semi-exponential model, the employed GLM model with less limitations and simple algorithm has a higher saturation level (nearly to 300 m^3^/ha) and higher sensitivity to high forest GSV values than the semi-exponential model. Furthermore, by reducing the external disturbance with the help of time average, the accuracy of estimated GSV is improved using fused polarimeric characteristics, and the estimation accuracy of forest GSV was improved as the images increase. Using the fused polarimetric characteristics (Dbl×Vol/Odd) and the GLM, the minimum RRMSE was reduced from 33.87% from single SAR image to 24.42% from the time series SAR images. It is implied that the GLM is more suitable for polarimetric characteristics derived from the time series SAR images and has more potential to improve the planted forest GSV.

## 1. Introduction

With the growing area of planted forests, it is of great significant to reduce the carbon dioxide emission and slow down the global warming, due to the decrease of natural forests. Understanding how to evaluate and well-manage planted forests is becoming increasingly important [1,2,3,4]. The growing stock volume (GSV), defined as the total stem volume of living trees, is a basic key indicator for monitoring the planted forest resource at regional scales [4,5,6]. Traditionally, the GSV is derived by measuring heights and diameters at breast height in ground plots. However, the GSV of samples can hardly be realized by traditional ground survey in mountainous regions, which is time-consuming, labor-intensive and costly [3,4]. Optical remote sensing techniques have been widely used to monitor the forest resource by establishing the relationship between in-situ GSV and characteristics derived from remote sensing images [5,6]. Various images from optical remote sensing have bands to discriminate the difference between the forest type and other types [5,7]. However, disturbed by clouds, fog and mist, the high-quality optical images are hardly acquired in mountain areas, which are the major distribution regions for the planted forest. Microwave remote sensing technology, which is less affected by weather conditions, has ability to measure the forest GSV by their penetration depth related to wavelength [8,9,10]. Moreover, the integration of synthetic aperture radar and polarimetric techniques, polarimeric property related to structural characteristics of forest canopy is detected and the polarimeric information derived from quad-polarimetric SAR images has more potentiality for monitoring the GSV in these areas [9,10,11,12].

Initially, the model describing the relationship between the forest GSV and polarimeric information is critical for forest GSV estimation [7,9,12]. Physical models have many parameters, and are too complex to estimate the forest GSV [13,14]. Empirical models, including the first-order linear model, multi-variable linear model and the linear model based on the allometric equation, are often employed to describe the relationships between backscattering coefficients and GSV [8,15,16]. However, they omit the scattering property related to forest structure parameters. Semi-empirical models, considering the backscattering of ground and forest, are also applied to estimate the stand-level GSV using X and C band SAR images [17,18,19,20]. Further studies found that the polarimetric characteristics obeyed the exponential distribution [19,20], and a semi-exponential model, derived from the simplified water cloud model (WCM) based on the exponential distribution, was proposed in expressing the interaction between polarimetric characteristics and GSV [21,22,23,24,25,26].

Compared with the physical models and empirical models, fewer parameters contain in the semi-exponential model and it is convenient to estimate the forest GSV using different polarimeric information, including backscattering coefficients and powers of scattering from polarimetric decomposition [23,25]. The extra advantage of the semi-exponential model is that the saturation level is considered as a parameter and solved by estimating GSV [7,21,27]. However, the non-linear algorithms, with global optimum and the initial values of parameters, are necessary to accurately estimate forest GSV. Moreover, because of the low saturation level [7,16], the semi-exponential model is insensitive to the high GSV retrieval. Therefore, models with less limitations and higher saturation should be considered for mapping the GSV of plantations.

Besides the model, the sensitivity of the polarimetric characteristics derived from quad-polarimetric SAR images also determines the accurate of mapping the forest GSV [7,27]. There are several kinds of polarimetric characteristics related to GSV [7,28,29,30]. The backscattering coefficients from dual-polarization or quad-polarization SAR images were directly employed to retrieve the forest GSV [18,23,31,32,33,34,35], and the obtained results are in good agreement with the measured GSV in boreal forests [18,23,31,32,33,34,35,36]. However, without the explicit physical meaning of scatterings, the backscattering coefficients are easier to reach the saturation value at low forest GSV [16]. As we know, the coherence of SAR images from different satellites (bands and polarization) has great potential to estimate the forest GSV because of rather sensitivity to the height of scatterings [22,23,37]. it is difficult to acquire appropriate polarimetric SAR images, by limiting to the spatial baseline, temporal baseline and environment factors, especially in the forest regions. Polarimetric decomposition, extracting physical parameters from SAR images, without any ground measurements, is another resource of polarimetric characteristics [13,14]. The powers of scattering derived from various polarimetric decomposition approaches describe the process of scattering, and have been showed explicit physical meaning related to the scatterings [28,29]. In the previous works, the approaches of model-based decompositions have been proposed for polarimetric decomposition, such as Freeman three-component decomposition, Yamaguchi four-component decomposition, Arii three-component decomposition and Neumann decomposition [28,29,38,39,40]. Recently, several investigations were undertaken to further improve the methods by including more sophisticated vegetation volume scattering model or by limiting the negative scattering power effects, such as 4-component Rotated decomposition, the generalized 4-component unitary decomposition and Stochastic Distance-based 4-component decomposition [40,41,42]. Furthermore, a hybrid model-based and eigenvalue/eigenvector-based polarimetric decomposition technique was also proposed to derive the powers of scattering in vegetation cover regions [43]. Due to their simplicity and computational ease, the Yamaguchi decomposition and its improved algorithms are widely applied to estimate forest GSV [13,36,39,40,41,42].

Moreover, previous studies have demonstrated the advantages of using time series SAR images for GSV estimation over single polarimetric SAR images [17,20,44,45,46]. The methods for estimating GSV from time series SAR images, such as the regression model, average, and weighted average, often take the estimated GSVs from single SAR images as independent variables [47,48,49]. However, the sensitivity between the polarimetric characteristics and the forest GSV is affected by external disturbance, and the GSVs estimated from single SAR images are also affected by the quality of polarimetric characteristics. In addition, the image acquisition intervals also have influences on the results. Large intervals, ranging from several months to two years, lead to great uncertainty in the forest scatterings [47,48]. Therefore, in order to reduce the influence of season on forest GSV estimation, the intervals of the acquired time series images should be considered.

In this study, to accurately retrieve the planted forest GSV, four ALOS PALSAR-2 polarimetric SAR images of the planted forest in Youxian, China, were acquired during the forest growing season with intervals ranging from 15 to 42 days. Five un-fused and fused ploarimetric characteristics were derived by the Yamaguchi decomposition from single and time series SAR images. Then, to overcome the disadvantage of semi-exponential model, we developed a general linear model (GLM) for mapping stand-level forest GSV using these un-fused and fused polarimetric characteristics. The sensitivity and accuracy of the new model in estimating the GSV, using time series polarimetric SAR images, were analyzed in our study area.

## 2. Study Area and Data

### 2.1. Study Area

The study area (10,122.6 ha) is located in Youxian County, Hunan province of China (Figure 1a). The plantation forest is the dominant in the forest farm, and the classification of forest farm is ecological welfare forest farm and the elevation varies from 115 m to 1270 m. The major tree species includes Chinese fir (Cunninghamia Lanceolata), Pinus massoniana Lamb, bamboo, Liriodendron chinense and Cinnamomum camphora, and the planted Chinese fir is the dominant species. The forest coverage is 86.24% with GSV was close to 879,705 m^3^ in 2014.

### 2.2. Ground Data

Based on the data from forest resource inventory and planning, most of the planted Chinese fir forests are located in the north and east part of the forest farm. Considering the age group and distribution of GSV in planted forest, 50 plots of the planted Chinese fir with a random stratification sampling were measured between 2016 and 2017 (Figure 1b). In all ground measured plots, the percentage of young, immature and mature forests is 10%, 52%, and 38%, respectively. The percentage of mixed sample plots was 12% and the other tree species in mixed plots were broad-leaved tree species (Liriodendron Chinese and Cinnamomun camphcra) and Masson pine, but the percentages of broad-leaved tree species and Masson pine were less than 8%. The size of plots (30 m × 30 m or 20 m × 20 m) was applied depending on the local terrain condition. Moreover, the corner points and central points of the investigated sample plots were surveyed using the global positioning system (GPS).

Totally, there were 4935 trees measured for the inventory in all 50 plots. Trees with the DBH smaller than 5 cm were not measured in the plots. Two parameters of each tree, the height and diameter at breast height (DBH), were measured to calculate the stem volume of each tree, then the GSV of plot was retrieved by [45,46],
(1)$$GSV=\sum g_{i}\times \left(H_{i}+3\right)\times f$$
where GSV is the stem volume of the plot, $g_{i}$ is the cross-section area of each tree related to measured DBH, and $H_{i}$ is the height of each tree, f is the trunk taper coefficient of the planted Chinese fir related to height and DBH [43,44]. In all plots, the relationship between the average height and DBH of each plot was illustrated in Figure 2b, the maximum average height and DBH is 20.5 m and 29.48 cm, respectively (Figure 2a). Moreover, the GSV is close to 63 m^3^/ha for young stands and the average GSV of over-mature forests is up to 322.59 m^3^/ha. The relationship between the GSV and measured DBH is illustrated in Figure 2b.

### 2.3. Polarimetric SAR Images and the Digital Elevation Model (DEM)

Table 1 shows the ALOS-2 PALSAR-2 L-band in full polarimetry (http://global.jaxa.jp/) used for estimating the GSV in this study. Four single look complex (SLC) quad-polarimetric L-band SAR (HH + HV + VH + HH) images at level 1.1 were acquired at 04:22 am from 30 June to 22 September, 2016 and the intervals of the acquired images range from 14 days to 42 days. The off-nadir angle is 38.99 degree on the descending orbit, and the pixel resolution in the azimuth direction and the slant range direction is 2.83 m, and 2.86 m, respectively (Figure 2a). In order to geocode the SAR images, ASTER GDEM and the slope of study area (Figure 2b) with a spatial resolution of 30 m was employed in the following data processing.

Based on the information from local weather forecast, the weather conditions of those four quad-polarimetric SAR images were different at the acquired time (Table 2). The wind direction of images acquired on 30 June and 14 July was south with less than 3 grades. Inversely, the wind direction is north and the grade was ranged from 3 to 4 for the images acquired on 25 August and 22 September. It was cloudy and clear at the acquired time on 30 June and 22 September, and showers for the images acquired on 14 July, and 25 August, respectively. Moreover, for the images acquired on 14 July and 25 August, it was always rainy for three days before acquired time and the moisture of ground was larger than that on 30 June and 22 September.

## 3. Method

### 3.1. Pre-Processing of the Polarimetric SAR Images

To retrieve the forest GSV, the polarimetrical calibration was performed to reduce the impact of Faraday rotation [5,38,41,47] Then, the errors induced by terrain slope were corrected by polarization orientation angle compensation and terrain radiometric correction with external digital elevation models (DEM) [5,49,50,51,52,53,54,55]. After that, the Lee filter (7 × 7) was adopted to retrieve homogeneous pixels and reduce the errors caused by the speckle noise [28]. Finally, the coherency matrix was formed by the spatially averaged algorithm, which could be used to extract the polarimetric characteristics in the subsequent processing [28,51].

### 3.2. Extraction of Time Series Polarimatric Characteristics

#### 3.2.1. Yamaguchi Decomposition and Polarimatric Characteristics

Polarimetric characteristics are commonly used to describe the scattering features related to the forest structure parameters, such as height, DBH and age of trees. Therefore, retrieving the polarimetric characteristics from the coherency matrix is a crucial step for forest GSV mapping.

Yamaguchi proposed a four-component polarimetric decomposition method, which can retrieve the power of surface scattering (Odd), double-bounce (Dbl), volume scattering (Vol) and Helix scattering (Hlx) [28,52] without using any ground measurements. Normally, the power of these scatterings is related to forest structure parameters, incidence angle, wavelength and terrain, and so on. The four sub-matrices derived from the coherency matrix have a relationship as follows,
(2)$$P_{t}=P_{Odd}+P_{Dbl}+P_{Vol}+P_{\mathrm{Hlx}}$$
where $P_{t}$ is the span of backscattering, $P_{Odd}$ is the component of the surface scattering from non-forest ground, $P_{Dbl}$ is the component of the double-bounce scattering from a dihedral corner reflector, such as tree trunks and the interaction between the trunks and big branches, $P_{Vol}$ is the component of the volume scattering from forest canopy with a number of randomly oriented dipoles, and $P_{Hlx}$ is the component of the Helix scattering related to buildings. The power of $P_{Odd}$, $P_{Dbl}$ and $P_{Vol}$ can be considered as un-fused polarimetric characteristics to estimate the forest GSV.

Some fused polarimetric characteristics formed by multiplication or division of polarimetric components are also used to map forest GSV, including $P_{Dbl/Odd}$, $P_{Vol/Odd}$, $P_{Dbl\times Vol}$ and $P_{Dbl\times Vol/Odd}$. The performances of the un-fused and fused polarimetric characteristics in forest GSV mapping would be analyzed in Section 4.2.

#### 3.2.2. Time Series Polarimetric Characteristics

The power of scatterings from different SAR images with a short interval could be quite different because of the disturbance of wind, soil moisture and speckle noise. However, the forest GSV cannot have great changes in a short interval, even months. Therefore, it is rather difficult to evaluate the reliability of the forest GSV retrieved from only one single SAR image.

Increasing the SAR images acquired in the same period, which are the time series images, is an effective way to guarantee the reliability of the mapped forest GSV. After pre-processing, the time series polarimetric characteristics, extracted by the Yamaguchi decomposition, were geocoded with external DEM and then registered with reference images. After that, the un-fused and fused polarimetric characteristics are calculated by the temporal average at the pixel scale. The fused polarimetric characteristics were constructed by fusing power of the surface scattering, double-bounce scattering and volume scattering. Due to the mismatches between the measured plots and pixels, the polarimetric characteristics of each plot were extracted by scale matching using the spatial average. Figure 3 is the flowchart of extracting the time series polarimetric characteristics.

### 3.3. The General Linear Model for GSV Estimation

Theoretically, the model for GSV estimation should describe the relationships between the polarimetric characteristics and forest GSV as simple as possible [13,23,26]. The polarimetric characteristics retrieved from the coherency matrix obey the exponential distribution. Therefore, we used the general linear model (GLM) to describe the relationships between polarimetric characteristics and forest GSV,
(3)$$\sigma =e^{a_{0}+a_{1}\times GSV}$$
where $\sigma_{}^{}$ denotes the power of polarimetric characteristics, GSV is the growing stock volume (m^3^/ha), $a_{0}$ and $a_{1}$ are the unknown parameters. To solve the unknown parameters, the non-linear model is reformed to a linear model with a link function,
(4)$$g(\sigma )=\ln(\sigma )=a_{0}+a_{1}\times GSV$$
where g(σ) is the logarithmic form of σ. The two unknown parameters, $a_{0}$ and $a_{1}$ are solved by the least square linear regression algorithm without setting initial values. The forest GSV can be calculated as follows:(5)$$GSV=\frac{\ln(\sigma )-a_{0}}{a_{1}}$$

In this study, the semi-exponential model proposed by Wagner et al. [36] was also employed as contrast model to estimate t forest GSV,
(6)$$\sigma_{}^{}=\beta_{s}+(\beta_{n}-\beta_{s})\cdot e^{\frac{GSV}{-k}}$$
where $\sigma_{}^{}$ is one of the selected polarimetric characteristics, GSV is the measured growing stock volume (m^3^/ha). $\beta_{n}$ refers to the polarimetric characteristic of non-vegetated area, and $\beta_{s}$ refers to those of the forests with the highest GSV. $k_{}$ is the saturation level of the forest GSV. $\beta_{n}$, $\beta_{s}$ and $k_{}$ are unknown parameters, whose initial values are determined by the range of polarimetric characteristics. The non-linear algorithm (software: Python) was employed to estimate the unknown parameters.

### 3.4. Model Assessment

To select the variables for the GSV estimation, Pearson correlation coefficient (γ) at the significant level of 0.01 was adopted to evaluate the relationship between the polarimetric characteristics and GSV. The approach of leave-one-out cross-validation (LOOCV) was used to compare the performance of selected models and variables, and the statistical criteria, such as the Root Means Square Error (RMSE), the coefficient of determination (R^2^) and the relative RMSE, are employed to assess the difference between the predicted forest GSV data and the observed data [16,21].

## 4. Results

### 4.1. The Extracted Polarimetric Characteristics

To match the size of the measured plots, un-fused and fused the polarimetric characteristics derived from the Yamaguchi polarimetric decomposition were spatially averaged with a size of 7 × 7. The relationships between the measured forest GSV and the polarimtric characteristics were plotted in Figure 4. Five time series formed by images 1, 2, 3 and 4, acquired on 30 June, 14 July, 25 August and 22 September, respectively and with different intervals (14 days to 84 days), were selected to investigate the influences of external disturbances. Pearson’s correlation coefficient between GSV and the selected characteristics were also calculated (Table 3). The surface scattering (Figure 4a) and double-bounce scattering (Figure 4b) are the major scatterings in the planted Chinese fir forest. Moreover, the surface scattering, double-bounce, the Dbl/Odd (Figure 4d) and Dbl×Vol/Odd (Figure 4g) obey the exponential distribution. But, there are some anomaly values between the forest GSV and the power of Vol/Odd (Figure 4e) and Dbl×Vol (Figure 4f).

For the un-fused polarimetric characteristics from single SAR images, some correlation coefficients of the volume scattering are smaller than the critical value of 0.345 at the significance level of 0.01, being too weak to be used for the forest GSV mapping. Although, the power of surface scattering has obvious negative correlation with the forest GSV (−0.557~−0.415), it is not rational to be viewed as an independent variable in the subsequence study as it relates to the ground without forest. The correlation coefficients of Dbl and Dbl×Vol/Odd have significant positive correlations ranging from 0.392 to 0.702, so they are considered as independent variables. The other fused characteristics, Dbl/Odd, Vol/Odd and Dbl×Vol, with significant negative correlation (−0.709~−0.367) are also considered as independent variables.

As Table 3 shows, the correlation coefficients of the un-fused and fused polarimetric characteristics from time series images were significantly higher than those of single images, except the combination of 2 and 3 (acquired on 14 July and 25 August). The Odd, Dbl/Odd, Vol/Odd and Dbl×Vol have negative correlations and the remainders have positive correlations. This is the same for both the single and time series images. Overall, all these characteristics have significant correlations with GSV and could be used as independent variables to estimate the forest GSV, except the power of surface scatting (Odd) and volume scattering (Vol).

### 4.2. Accuracy of GSV Estimation Using Single SAR Images

Before retrieving the model parameters, the measured plots in the shadow region were discarded. Using the five independent variables (Dbl, Dbl/Odd, Vol/Odd, Dbl×Vol and Dbl×Vol/Odd) derived from each single SAR image, the parameters of GLM were solved by the least square algorithm. Moreover, to compare the results of the semi-exponential model, the optimal solutions of the semi-exponential model were also solved by the non-linear solution algorithm and the proposed initial value of unknown parameters. The LOOCV method was employed to calculate the RMSE and RRMSE between the estimated and measured GSV (Figure 5).

For the results of the GLM model, the RMSE is between 69.01 m^3^/ha and 83.98 m^3^/ha and the RRMSE is between 32.60% and 39.22%. The minimum and the maximum RMSE of the estimated GSV were derived from the double-bounce scattering (Figure 5a) acquired on 30 June 2016 and Vol/Odd (Figure 5c) acquired on 22 September 2016, respectively. Furthermore, the RMSE of the GSV estimated by the Dbl/Odd (average values = 73.44 m^3^/ha) is much smaller than that estimated by other independent variables. For the semi-exponential model, the RMSE is between 63.73 m^3^/ha and 93.01 m^3^/ha, and the RRMSE is between 31.97% and 45.67%, which are larger than those of the GLM. Moreover, the estimated GSV using GLM were compared with those using the semi-exponential model, and there is significant difference between them by the differences in the significant test at the significant level of 5%.

The fitted curves for images acquired on 30 June and 22 September were plotted in Figure 6, which illustrates the forest GSV estimation capability of the GLM model and the semi-exponential model. GLM was more sensitive than the semi-exponential model for high GSV values, even for the forest GSV larger than 300 m^3^/ha. The semi-exponential model lacks the capability to estimate the GSV when the forest GSV is larger than 100 m^3^/ha. The semi-exponential model also has a saturation level lower than that of GLM. The trends of fitted curves are more clearly for the polarimetric characteristics from the image acquired on 25 August 2016 (Figure 6f–j).

To further analyze the sensitivities of the two models, the coefficients of determination (R^2^) between the measured and the estimated GSV with five independent variables from a single image were calculated (Table 4). The R^2^ of GLM is larger than that of the semi-exponential model for most independent variables. For example, the R^2^ of Dbl ranges from 0.43 to 0.60 for GLM, and from 0.46 to 0.58 for the semi-exponential model. Figure 7 illustrates the scatter diagrams between the observed and estimated GSV using the single SAR image acquired on 30 June 2016. Obviously, the GSV were overestimated by the semi-exponential model with Dbl (Figure 7f), Vol/Odd (Figure 7h) and Dbl×Vol/Odd (Figure 7j) and partially underestimated by GLM. In general, GLM is superior to the semi-exponential model using the same un-fused and fused independent variables.

### 4.3. Accuracy of GSV Estimation Using Time Series SAR Images

By the temporal average approach, polarimetric characteristics were derived from time series with different intervals and taken as independent variables to estimate the forest GSV. The estimated results of GLM and the semi-exponential model are compared in Table 5. The RMSE ranges from 59.21 m^3^/ha to 70.76 m^3^/ha for the semi-exponential model, and from 50.64 m^3^/ha to 70.49 m^3^/ha for GLM. Compared with the results of the Dbl×Vol/Odd from single SAR images, the time series images can significantly improve the estimation accuracy. However, using the semi-exponential model, the fused polarimetric characteristics, derived from the images acquired on 14 July and 25 August, induced large errors and more than 25% plots with errors exceeding the threshold. Moreover, the minimum RRMSE (24.42%) is observed by GLM when all the time series images are used. Furthermore, the RMSE and RRMSE decrease as the number of SAR images increases (Figure 5), except that of the independent variables from the time series images (2 and 3 in Table 5).

Figure 8 compares the estimation accuracy of GLM and the semi-exponential model using time series images. For un-fused and fused independent variables, the estimated GSV using the semi-exponential model easily reaches the saturation point, which is smaller than 200 m^3^/ha for some polarimetric characteristics. On the contrary, the GLM is more sensitive to the high GSV because of the higher saturation levels. In addition, using the GLM, the forest GSV estimated, using the independent variables from time series images, is more stable and more accurate than that using the variables from single SAR images (Figure 6).

Figure 5 illustrates the RMSE and RRMSE of the time series polarimetric characteristics got by the semi-exponential model and GLM. For each selected independent variable, the RMSE and RRMSE of the time series images are smaller than that of single SAR images. The average RRMSE is about 35.36% for the two models using the power of double-bounce scattering (Dbl) from single SAR image (Figure 5f). With the power of the double-bounce from time series images, the RRMSE decreases to 30.26% and 32.53% for GLM and the semi-exponential model, respectively. The improvements are more significant for Dbl/Odd (Figure 5g), Vol/Odd (Figure 5h) and Dbl×Vol/Odd (Figure 5i). Moreover, for the GLM results, the RRMSE of all variables from time series images is smaller than 30%, and that of variables from single images is larger than 32%.

Furthermore, the coefficients of determination (R^2^) between the measured and the estimated GSV using variables from time series images are listed in Table 6. For each time series combination, the R^2^ of GLM is slightly larger than that of the semi-exponential model. The maximum R^2^ 0.71 is got by GLM with Dbl×Vol/Odd. Therefore, the polarimetric characteristics with time series average can increase the estimation accuracy of GLM.

The scattering diagrams between the measured and the estimated GSV (Figure 9) show that the results of GLM have less overestimated or underestimated forest GSV than that of the semi-exponential model. Particularly, in the results of the semi-exponential model, some plots with high GSV induce larger errors and the estimated GSV exceed the rational range. Therefore, we mapped the GSV of the planted Chinese fir forest by GLM using the fused polarimetric characteristics (Figure 10), and the estimated forest GSV of most regions ranges from 100 m^3^/ha to 450 m^3^/ha.

## 5. Discussion

### 5.1. The Sensitivity of Polarimetric Characteristics

Normally, the bands of SAR image and the polarimetric decomposition method are the major factors for assessing the sensitivity between polarimetric characteristics and forest GSV. As we know, lower frequencies (L-band and P-band) are more suitable than higher frequencies (X-band and C-band) as saturation emerges at higher forest growing stem volume levels [9,10,11]. That is to say, after a specific GSV level, the further increase of GSV caused no further increase in the intensity of polarimetric characteristics. Moreover, several decomposition methods, such as the Cloude decomposition, Pauli decomposition, Freeman three-component decomposition, and Yamaguchi four-component decomposition were applied for estimating the forest GSV in the previous study [9,16,25]. The improved decomposition methods, such as 4-component Rotated decomposition, the generalized 4-component unitary decomposition and Stochastic Distance-based 4-component decomposition are proposed to improve the powers of scatters [6,55]. The results showed that the Yamaguchi decomposition method is more suitable for mapping forest GSV.

In our study, the Yamaguchi four-component decomposition method was employed to retrieve the polarimetric characteristics, and the results showed that the measured GSV has strong positive correlations (ranged from 0.392 to 0.702), with the power of double-bounce scattering, and strong negative correlations (ranged from −0.613 to −0.415) with the power of surface scattering (Figure 5a,b and Table 3). However, the power of volume scattering related to forest canopy should be sensitive to the GSV, theoretically, but it showed have weak sensitivity to GSV for the un-fused polarimetric characteristics in our study. The reason is that the powers of radar backscattering at L bands is dominated by the scattering process from the trunks and large branches, while powers of X and C band is dominated by the scattering process in the crown layer of small branches, needles and therefore, it does not penetrate and scatter significantly from the stem.

Compared with un-fused polarimetric characteristics, the fused polarimetric characteristics can be improved the sensitivity to GSV (Table 3). Due to the complexity of the forest structures, polarimetric characteristics related to single scattering property is insufficient in describing the forest GSV. In this study, the average values of the Pearson correlation coefficients of single scattering (Odd, Dbl and Vol) are −0.469, 0.511 and 0.399 before the combination, respectively. After the combination, the Pearson correlation coefficients of fused polarimetric characteristics (Dbl/Odd, Vol/Odd, Dbl×Vol, and Dbl×Vol/Odd) are up to −0.476, −0.519, −0.510 and 0.507, respectively. The improvement in the fused polarimetric characteristics was caused by accurate description of forest structures using multi-scatterings. Fused by other scatterings, the power of volume scattering with weak correlation with GSV was also helpful to enhances the sensitivity, such as Vol/Odd and Dbl×Vol.

Moreover, sensitivity between polarimetric characteristics and forest GSV are also affected by weather conditions at the acquired time. In this study, the Pearson correlation coefficients of images acquired on 14 July and 25 August are obvious lower than that derived from the images acquired on 30 June and 25 September (Table 3). Based on the weather information listed in Table 2, the discrepancy of the Pearson correlation coefficients was mainly induced by the moisture of forest and ground at the acquired time. It was always rainy for three days for the images acquired on 14 July and 25 August (Table 3). Additionally, the sensitivity of polarimetric characteristics is also slight influenced by the wind force scale, and the Pearson correlation coefficients derived from the image acquired on 30 June are significantly higher than that from the images acquired on 25 September.

### 5.2. The Estimation Accuracy of GLM

In the previous study, the semi-exponential model was applied to estimate the forest GSV using different polarimetric characteristics and the values of RRMSE were ranged from 25% to 65% [7,15,21,22,26]. However, the parameters of the semi-exponential model should be solved by complicated algorithms with given appropriate initial values, which are not always available due to the lower accuracy observations. In our study, with simple forms and reliable algorithm, the GLM was employed to estimate the GSV of the planted Chinese fir forest. The parameters of GLM can be easily solved by the simple least square algorithm without initial values. For the powers of double-bounce derived from time series image (Figure 6), the improvement was observed in estimated results between the GLM (RRMSE: 25.84–32.77%) and the semi-exponential model (RRMSE: 28.99–35.34%). The improvement in the results from the GLM was also observed for the fused polarimetric characteristics, such as Dbl/Odd and Vol/Odd. Therefore, it was proven that the GLM is able to estimate the GSV of planted forest.

For the time series polarimetric characteristics with different intervals, significant improvement of estimated planted forest GSV was observed using the GLM. The RRMS and RRMSE between the measured and the estimated GSV by GLM are much smaller than that obtained by the semi-exponential model for both the un-fused and fused independent variables (Figure 6). In particular, for the fused polarimetric characteristic (Dbl×Vol/Odd), the values of RRMSE (Table 6) range from 24.42% to 33.46% using the GLM, and from 29.60%, to 36.29%, using the semi-exponential model, respectively. The sensitivity and stability are obvious improvement for the fused independent variables, including Dbl/Odd (Figure 6g), Vol/Odd (Figure 6h) and Dbl×Vol/Odd (Figure 6j). Compared with the results from single SAR image, independent variables from time series SAR images have more potential to map the planted forest GSV.

Moreover, the saturation level of GSV is a key indicator for assessing the performance of the estimation model. According to the fitted curves from single and time series polarimetric characteristics (Figure 6 and Figure 8), the saturation level of GLM is higher than the semi-exponential model, for either the fused or un-fused polarimetric characteristics. Once the GSV is larger than 300 m^3^/ha, the semi-exponential model cannot accurately retrieve the GSV, and low saturation levels may also cause overestimated GSV by the semi-exponential model (Figure 7 and Figure 9). Therefore, GLM is more potential than the semi-exponential model in estimating GSV.

### 5.3. Single and Time Series SAR Images

The acquisition seasons and intervals of the polarimetric SAR images are rather important for time-series processing, as they lead to different interactions between trees and microwaves. Using the polarimetric characteristics, derived from single SAR images, cannot reduce the external disturbance [20,39,46,47]. In this study, four images were acquired in the growing season, which is shorter than three months (Table 1), and the discrepancies of sensitivity between these images were induced by external factors (Figure 5 and Table 3), including wind, ground moisture and speckle noise. For the single SAR image, the moisture and wind force scale major affected the sensitivity of the polarimetric characteristics (Table 3) and estimated forest GSV. It is unreliable to estimate the forest GSV using a single SAR images.

For the time-series processing, the normal approach is to estimate the GSV from each single SAR image, and then apply average, weighted average or complicated time-spatial models, to map the GSV using multi-SAR images [20,44,47,48,49]. Also, we can directly derive polarimetric characteristics from time-series SAR images with time-average, which are then used to estimate the forest GSV. By reducing the external disturbance, the polarimetric characteristics with less noise can be derive from time series SAR images using the latter approach. In this study, we chose the latter one. As showed in Table 3, the independent variables were derived from the time series images with different intervals, and have much higher Pearson correlation coefficients, than those of the variables derived from single SAR images. In addition, for the un-fused and fused polarimetric characteristics, the difference of the Pearson correlation coefficients between each time series is smaller than that from single SAR image. It is confirmed that polarimetric characteristics from time-series SAR images have less noise by time average.

Moreover, employed the semi-exponential model or the GLM, the values of RMSE and RRMSE were obviously decreased after using the variables derived from the time series SAR images. For the fused polarimetric characteristics (Dbl×Vol/Odd), the minimum RRMSE was 36.97% for the semi-exponential model and 33.87% for GLM from single SAR image, and these values were reduced to 29.60% and 24.42% from the time series SAR image (Table 6).

## 6. Conclusions

This study proposed the GLM for the stand-level forest GSV estimation using the time series L-band ALOS PALSAR-2 quad-polarimetric SAR images. Before the estimation, the un-fused and fused polarimetric characteristics were derived by Yamaguchi decomposition and the polarization response was analyzed with the GSV in the planted Chinese fir forest. The main results of this research are as follows:

(1) As the planted Chinese fir forest has narrow and short canopies with many canopy gaps, the measured GSV has strong positive correlations with the power of double-bounce scattering and weak correlations with the power of volume scattering. Furthermore, the power of the fused polarimetric characteristics is more sensitive to forest GSV than the power of un-fused polarimetric characteristics.

(2) The results showed that the RRMSE of the GLM results are much smaller than that of the semi-exponential model results. For the forest with high GSV, GLM has higher reliability than the semi-exponential model.

(3) The independent variables derived from time series polarimetric SAR images are more sensitive to forest GSV, and the minimal RRMSE, about 24%, was observed by Dbl×Vol/Odd using the fused independent variables from time series.

In summary, the proposed general linear model has potential to describe the relationship between the forest GSV and time series polarimetric characteristics. In the future, the research will concentrate on improving the GSV estimation accuracy using multi-band and multi time images.

## Figures and Tables

**Figure 1 sensors-20-03957-f001:**
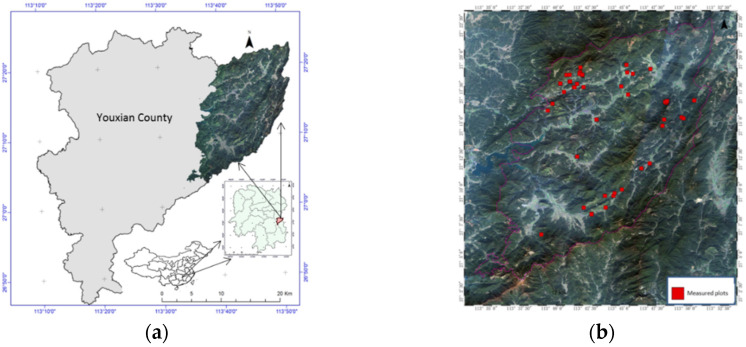
(**a**) Location of the study area; and (**b**) distribution of the ground measured plots (red squares) in the study area.

**Figure 2 sensors-20-03957-f002:**
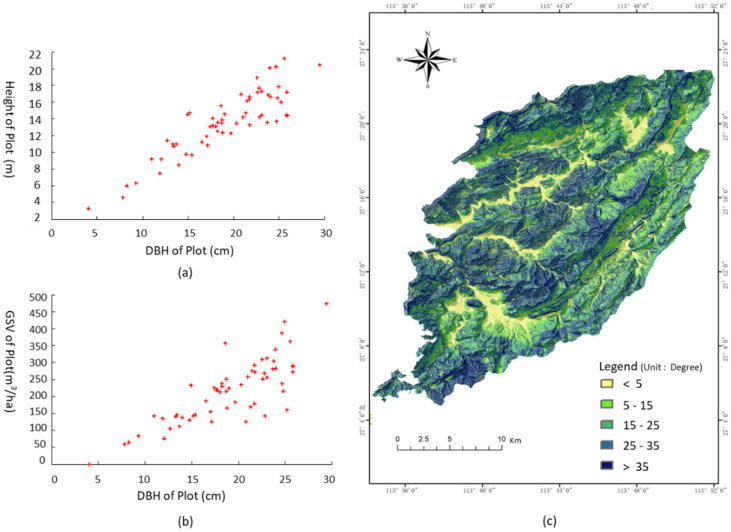
Scatterplot of measured forest parameters and the slope of the study. (**a**) The relationship between DBH and height; and (**b**) the relationship between DBH and forest GSV; (**c**) the slope of the study area derived from ASTER GDEM.

**Figure 3 sensors-20-03957-f003:**
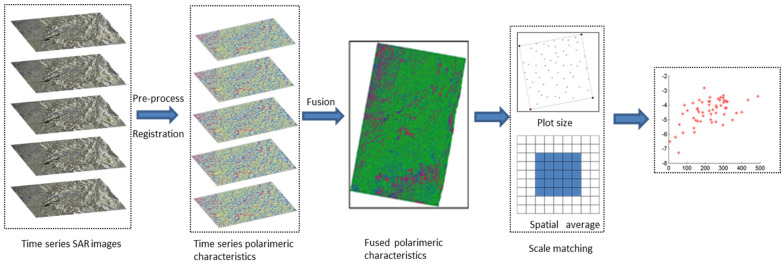
Flowchart of extracting polarimetric characteristics from the time series SAR images.

**Figure 4 sensors-20-03957-f004:**
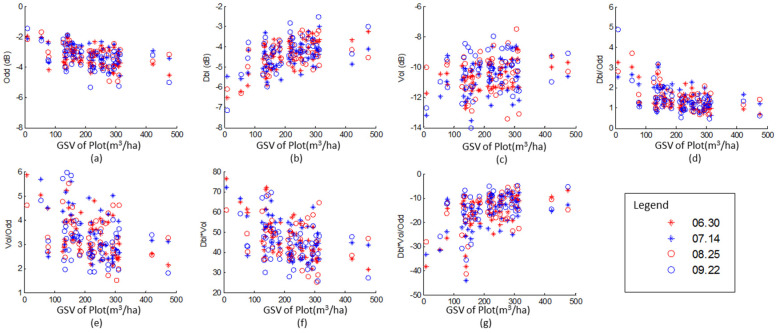
Scatter diagram of GSV and polarimetric variables obtained by the Yamaguchi decomposition from employed four SAR images. (**a**) surface scattering (Odd), (**b**) double-bounce scattering (Dbl), (**c**) volume scattering (Vol), (**d**) Dbl/Odd, (**e**) Vol/Odd, (**f**) Dbl×Vol and (**g**) and Dbl×Vol/Odd.

**Figure 5 sensors-20-03957-f005:**
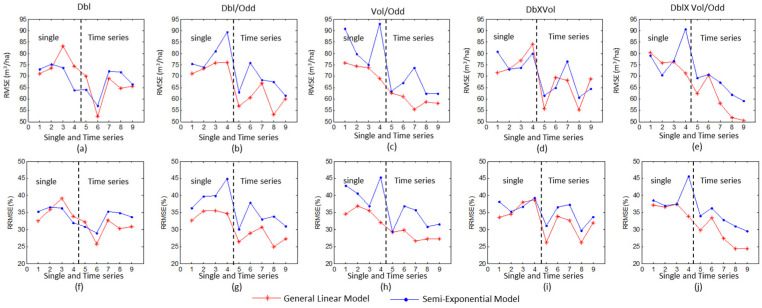
The RMSE (**a**–**e**) and RRMSE (**f**–**j**) between the measured and the estimated GSV retrieved by the selected independent variables using the GLM and the semi-exponential model. The numbers 1, 2, 3 and 4 in the X axis denote each single SAR image, and 5 to 9 denote the time-series.

**Figure 6 sensors-20-03957-f006:**
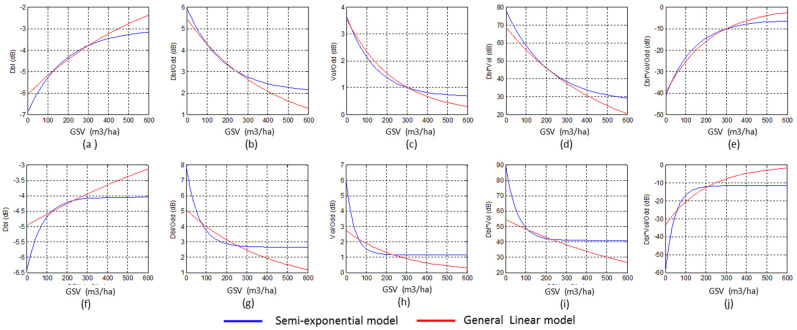
The fitted curves estimated by GLM and the semi-exponential model from single SAR image, (**a**–**e**) using the polarimetric characteristics from the image acquired on 30 June 2016 and (**f**–**j**) the image acquired on 22 September 2016.

**Figure 7 sensors-20-03957-f007:**
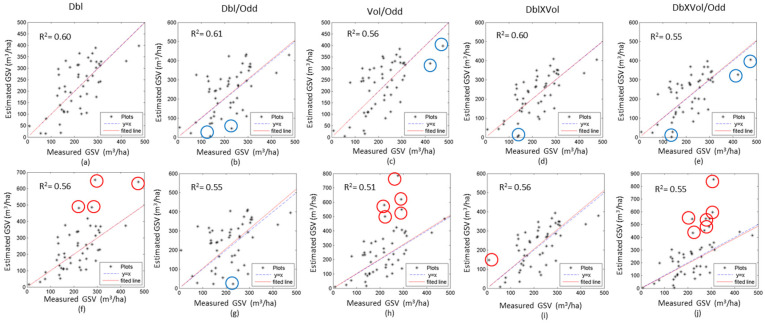
The scatter diagrams between the observed and estimated GSV using the SAR image acquired on 30 June 2016. The results of the GLM (**a**–**e**) and the semi-exponential model (**f**–**j**) are shown with difference variables. Red circle and blue circle denote the overestimated, and underestimated GSV, respectively.

**Figure 8 sensors-20-03957-f008:**
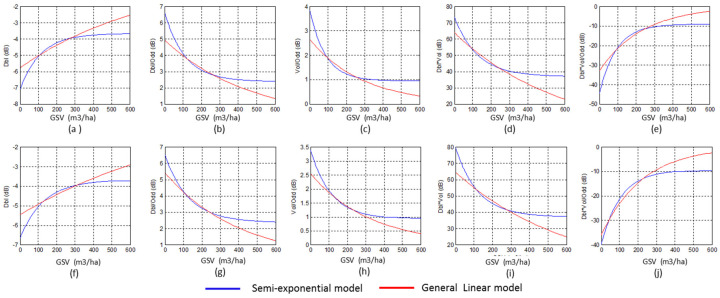
Comparison between the GSV estimated by GLM and the semi-exponential model using the variables from (**a**–**e**) time series images acquired on 30 June and 22 September 2016 (1, 4 in Table 5), and (**f**–**j**) time series images acquired on 30 June, 14 July, 25 August and 22 September 2016 (1, 2, 3, 4 in Table 5).

**Figure 9 sensors-20-03957-f009:**
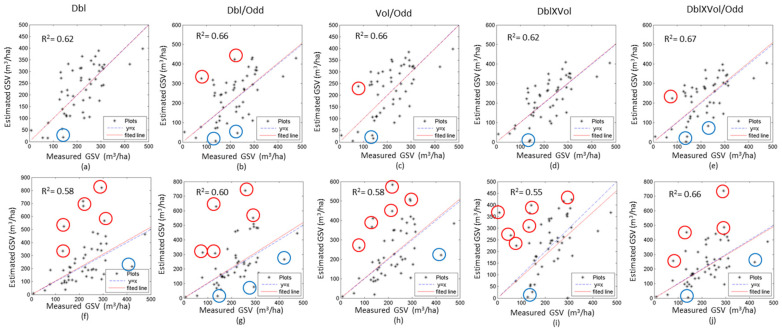
The scatter diagrams between the observed and the estimated GSV of the plots with the time series SAR images (1, 2, 3 and 4). The results using GLM (a–e) and semi-exponential model (f–j) are shown with difference variables. Red circle and blue circle denote over-estimated and under-estimated GSV, respectively.

**Figure 10 sensors-20-03957-f010:**
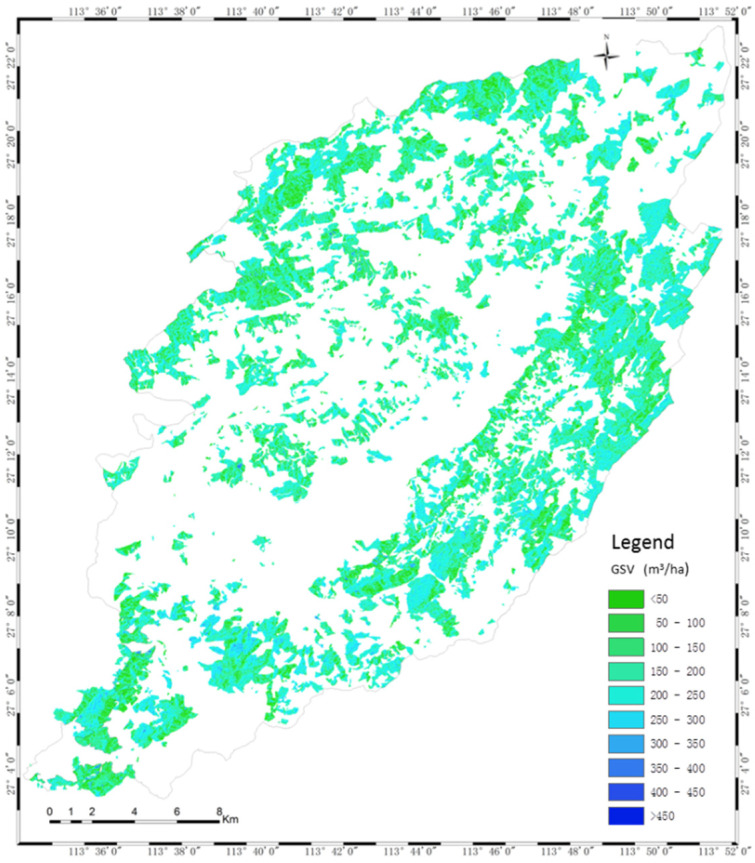
The GSV of the planted Chinese fir forests estimated by GLM using the Dbl×Vol/Odd from the time series image.

**Table 1 sensors-20-03957-t001:** Information of the polarimetric SAR images.

Sensor	Acquired Date	Central Latitude (Degree)	Central Longitude (Degree)	Band	Polarimetry
PALSAR-2	2016.06.30	27.179	113.676	L	HH + HV + VH + VV
PALSAR-2	2016.07.14	27.179	113.677	L	HH + HV + VH + VV
PALSAR-2	2016.08.25	27.179	113.678	L	HH + HV + VH + VV
PALSAR-2	2016.09.22	27.175	113.678	L	HH + HV + VH + VV

**Table 2 sensors-20-03957-t002:** Weather conditions of the polarimetric SAR images at acquired time and 3 days before acquired date.

Acquired Date	Temperature (°C)	Wind Direction	Beaufort Scale (Grade)	Weather at Acquired Time	3 Days Before Acquired
2016.06.30	26–35	South	less than 3	cloudy	sunny
2016.07.14	27–35	South	less than 3	Showers	Showers
2016.08.25	25–37	North	3–4	Showers	Showers
2016.09.22	19–30	North	3–4	clear	sunny

**Table 3 sensors-20-03957-t003:** Pearson’s correlation coefficients between the powers of polarimetric characteristics and forest GSV.

	Odd	Dbl	Vol	Dbl/Odd	Vol/Odd	Dbl×Vol	Dbl×Vol/Odd	Notes
06.30 (1)	−0.557	0.702	0.437	−0.569	−0.626	−0.709	0.624	Single
07.14 (2)	−0.421	0.392	0.360	−0.473	−0.462	−0.479	0.484	Single
08.25 (3)	−0.415	0.421	0.210 *	−0.385	−0.461	−0.367	0.415	Single
09.22 (4)	−0.484	0.529	0.207 *	−0.476	−0.526	−0.483	0.503	Single
1, 4	−0.612	0.696	0.416	−0.588	−0.625	−0.673	0.612	Time series
2, 3	−0.465	0.466	0.326 *	−0.467	−0.505	−0.466	0.487	Time series
1, 2, 3	−0.566	0.623	0.421	−0.550	−0.596	−0.609	0.578	Time series
2, 3, 4	−0.613	0.659	0.478	−0.590	−0.620	−0.655	0.607	Time series
1, 2, 3, 4	−0.594	0.634	0.436	−0.568	−0.608	−0.617	0.590	Time series

Note: * indicating that the coefficients are not significantly different from zero at the risk level of 0.01; Odd, Dbl and Vol indicate surface scattering, double-bounce, and volume scattering, respectively; 1, 2, 3 and 4, denote the images acquired on 30 June, 14 July, 25 August and 22 September, respectively.

**Table 4 sensors-20-03957-t004:** The coefficient of determination (R^2^) between the measured and the estimated plots GSV obtained by GLM and semi-exponential model using five un-fused and fused polarimetric characteristics from single SAR images.

Method	Acquired Date	Dbl	Dbl/Odd	Vol/Odd	Dbl×Vol	Dbl×Vol/Odd
General linear model	06.30	0.60	0.61	0.56	0.60	0.55
	07.14	0.58	0.51	0.51	0.56	0.51
	08.25	0.43	0.54	0.53	0.59	0.50
	09.22	0.52	0.53	0.58	0.55	0.57
Semi-exponential model	06.30	0.56	0.55	0.51	0.56	0.55
	07.14	0.58	0.50	0.53	0.54	0.52
	08.25	0.46	0.50	0.53	0.56	0.55
	09.22	0.54	0.36	0.33	0.41	0.32

**Table 5 sensors-20-03957-t005:** The GSV estimated by the semi-exponential model and GLM using Dbl×Vol/Odd.

		Semi-Exponential Model			General Linear Model		
		RMSE (m^3^/ha)	RRMSE (%)	R^2^	RMSE (m^3^/ha)	RRMSE (%)	R^2^
Dbl×Vol/Odd from single image	06.30 (1)	79.12	38.64	0.59	80.34	40.09	0.47
	07.14 (2)	70.50 *	36.97 *	0.65 *	75.92	36.60	0.46
	08.25 (3)	76.68 *	37.50 *	0.42 *	76.24	37.39	0.36
	09.22 (4)	90.68	45.67	0.21	71.23	33.87	0.47
Dbl×Vol/Odd from time series images	1, 4	69.27	34.05	0.50	62.23	29.80	0.53
	2, 3	70.76 *	36.29 *	0.42 *	70.49	33.46	0.45
	1, 2, 3	67.33	32.88	0.52	58.08	27.37	0.55
	2, 3, 4	61.80	31.04	0.58	52.88	25.32	0.63
	1, 2, 3, 4	59.21 *	29.60 *	0.55 *	50.64	24.42	0.60

Note: * indicates the percentage of plots with errors exceeding the threshold is than 25%.

**Table 6 sensors-20-03957-t006:** The coefficients of determination (R^2^) between the measured GSV and the estimated plots GSV using GLM and the semi-exponential model using five un-fused and fused polarimetric characteristics from time series SAR images.

Method	Combination	Dbl	Dbl/Odd	Vol/Odd	Dbl×Vol	Dbl×Vol/Odd
General linear model	1, 4	0.61	0.62	0.62	0.69	0.63
	2, 3	0.67	0.62	0.59	0.59	0.57
	1, 2, 3	0.59	0.64	0.63	0.59	0.65
	2, 3, 4	0.62	0.66	0.65	0.69	0.71
	1, 2, 3, 4	0.62	0.66	0.66	0.62	0.67
Semi-exponential model	1, 4	0.63	0.63	0.65	0.54	0.55
	2, 3	0.64	0.35	0.49	0.53	0.55
	1, 2, 3	0.59	0.61	0.56	0.58	0.63
	2, 3, 4	0.59	0.60	0.64	0.60	0.63
	1, 2, 3, 4	0.58	0.60	0.58	0.55	0.66

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
