# Peer review of "Estimating the Growing Stem Volume of the Planted Forest Using the General Linear Model and Time Series Quad-Polarimetric SAR Images"

_sensors, 2020, doi:10.3390/s20143957_

Round 1

Reviewer 1 Report

This paper used four L band ALOS-2 PolSAR data to extract the forest growing stem volume based on polarimetric characteristics and GLM of exponential distribution. The proposed method improves the accuracy of planted forest GSV. However, there still has some problems as follows:

The major errors:

(1) The Yamaguchi decomposition method is good for extracting polarimetric variable. How about the recently decomposition methods in forest GSV estimation. Some latest decomposition method should be discussed in this paper.

(2) The time series information is used for fusing the different temporal polarimetric information in experiment. And the experiment just compared the un-fused and fused data. In general, the more information used, the better results gained. So the different polarimetric decomposition method with un-fused and fused information should be added in the contrast experiment.

(3) The fusion strategy is Dbl/odd, Vol/odd, Dbl*Vol and Dbl*Vol/odd. Please give the reason why chosen these four fusion strategy and theoretical basis. Moreover, how about efficiency of other combinations?

(4) In general, for retrieving the GSV, the short band SAR is more suitable. The discussion part should be added the different bands in retrieving the GSV and analyzed the advantage of different bands, especially L band.  

(5) There are many careless errors of article format in abstract, equations, references et.al. The authors should be check and modify these problems.

Reviewer 2 Report

The study describes utilization of quad-polarimetric time series SAR images for estimation of growing stock of plantation forest in China. In general I can say the study is well written, the story of paper is easy to follow and the conclusion are in accordance with observation. I am convinced the paper can be accepted for publication just a minor comment:

  • on row 138 it is mentioned that the mixed stands are composed of broad leave species and fir (but it is not clear which broadleaves are there) also for the people unaware of such species it would be good to mention whether this species are always with leaves or they loose the leaves during winter or so, as this might influence slightly the measurements.
  •  the use of semi exponential model is described but I could not find description of algorithm (or better the software used to fit the model). Different software provide different quality of fit for non linear models please provide more description for the readers.

Reviewer 3 Report

The submission is nicely written and well detailed. From the conceptualization to the conclusions I am pleased with current content. SAR is a powerful tool to explore the environment and the proposed topic is a good application. I have only minor remarks:

  • is the Pearson correlation is the best tool for accuracy assessment, RMSE itself is a better tool (however, LOOCV is sensitive for outliers)
  •  the discussion should contain more comparisons of the results with the previous studies (even if this topic is unique, there were trying to determine the GSV or other characteristics of trees)
  • although the manuscript is well written, I have seen minor language issues (e.g. L447 a sentence begins with "and") so I suggest a thorough proofreading 

Round 2

Reviewer 1 Report

The authors replied all my question. Moreover, the part of introduction must be improved and added some latest decompostion method. 
